# Apparent Vertical Ionospheric Drift: A Comparative Assessment of Digisonde and Ionogram-Based Methods

Daniel Kouba<sup>1</sup>, Zbyšek Mošna<sup>1</sup>, Petra Koucká Knížová<sup>1</sup>

Department of Ionosphere and Aeronomy, Institute of Atmospheric Physics, Czech Academy of Sciences, Boční II/1401, 14100 Prague, Czech Republic

Correspondence to: Daniel Kouba (kouba@ufa.cas.cz)

Abstract. Reliable estimation of vertical plasma drift in the ionosphere is crucial for interpreting ionospheric dynamics and enhancing the accuracy of space weather models. This study provides a comparative assessment of direct Digisonde Drift Measurements (DDM) and indirect ionogram-based methods using parameters such as hmF2, h'F2, h'(3.5 MHz), and h'(0.8foF2). Two high cadence measurement campaigns were conducted at the mid-latitude observatory in Pruhonice, Czech Republic, during different phases of the solar cycle. The analysis focuses on evaluating measurement consistency, temporal coherence, and the influence of sampling step and averaging strategy on drift estimation. While DDM yields stable and robust results even at one-minute resolution, ionogram-derived methods are strongly affected by measurement uncertainty and ambiguity in virtual height interpretation—particularly at short time scales. However, at night, all methods converge when a 15-minute time interval is consistently applied both as the computation step and for subsequent smoothing. Under these conditions, coherent wave-like features in the vertical drift are reliably captured. The study outlines the strengths and limitations of each technique and provides recommendations for optimizing temporal resolution in ionospheric drift measurements, supporting improved methodology for future observational campaigns and model validation.

#### 20 1 Introduction

The investigation of plasma drifts within the ionosphere is crucial for understanding the complex dynamics of the Earth's upper atmosphere. In this weakly ionized plasma, the behavior of the neutral component is inseparably linked with that of the charged particles, especially at lower altitudes where collisions are frequent. With increasing altitude, the role of collisions decreases, and electromagnetic forces become dominant. A variety of forces—electric, magnetic, gravitational—alongside neutral winds and atmospheric pressure gradients govern the magnitude and direction of plasma motion, which can vary significantly across different locations and altitudes.

Plasma drifts are typically characterized by three vector components: northward, eastward, and vertical. Understanding all three components is essential for deciphering plasma transport, electron density fluctuations, and coupling between different regions of the ionosphere and thermosphere. Precise knowledge of drift velocities is essential for modeling space weather phenomena and their impact on communication and navigation systems.

A wide spectrum of methods, both observational and theoretical, has been developed to quantify drift velocities. These include satellite and rocket in-situ measurements, ground-based instruments such as ionosondes and Incoherent Scatter Radars (ISR), and computational simulations. Each approach offers unique advantages but also faces specific limitations. For instance, while in-situ rocket and satellite data provide valuable snapshots, they lack continuous long-term coverage. Ground-based techniques, on the other hand, can offer high temporal resolution but are spatially limited.

Importantly, this study does not aim to provide an exhaustive review of all existing knowledge about ionospheric drifts. Moreover, it is essential to note that even the term "drift" can be misleading in certain contexts. The vertical component of measured drift often does not correspond to the physical movement of plasma particles due to electromagnetic forces (i.e., the true plasma drift). Instead, what is sometimes detected is an apparent drift, where Doppler shifts arise from changes in the height of the ionospheric reflection layer due to ionization and recombination processes—especially during daytime. This can falsely suggest vertical plasma motion when, in reality, the reflective surface itself is shifting.

To study actual plasma motion, it is necessary to identify and exclude data influenced by such effects. Accordingly, some authors (e.g. Bittencourt and Abdu, 1981) limit their analyses to specific times and altitudes where these height variations are minimal, ensuring that the derived vertical drift values genuinely represent plasma motion. Nevertheless, even the data reflecting apparent drifts can be useful, for example, in identifying Travelling Ionospheric Disturbances (TIDs) and estimating their characteristics such as size, propagation speed, and direction.

Given these considerations, we find it beneficial to compare data obtained using different methods as comprehensively as possible. Even imperfect data, if correctly interpreted, can contribute significantly to the understanding of ionospheric processes.

# 50 1.1 Satellite and Rocket In-Situ Measurements

Satellite missions such as ROCSAT-1 have contributed significantly to our understanding of ionospheric drifts. During the active phase of Solar Cycle 23, ROCSAT-1's Ionospheric Plasma and Electrodynamics Instrument (IPEI) provided valuable data on ion density, temperature, and drift velocity. Fejer et al. (2008) used these measurements to develop a global empirical model for vertical drifts under moderate to high solar activity conditions, revealing strong longitudinal dependencies previously underappreciated.

The Communication/Navigation Outage Forecasting System (C/NOFS) satellite has also been instrumental in exploring equatorial plasma dynamics during solar minima (Pfaff et al., 2010; Kelley et al., 2014; Huang and Hairston., 2015). In addition, the ICON mission's Ion Velocity Meter (IVM) provides high-cadence in-situ measurements of ion drift velocity (including the vertical component near the magnetic equator) and has revealed seasonal, longitudinal and local time variations in both vertical and zonal drifts. (e.g., Heelis et al. 2017; Park et al. 2021).

#### 1.2 Incoherent Scatter Radars (ISR)

Among ground-based methods, the Incoherent Scatter Radar (ISR) is a powerful tool. The Jicamarca Radio Observatory near the magnetic equator has been a cornerstone in measuring both vertical and zonal components of plasma drift for decades (Rodrigues et al., 2013). Data from Jicamarca and the AE-E satellite have formed the basis for empirical models under varying seasonal and solar conditions (Scherliess & Fejer, 1999; Woodman et al., 2006; Fejer, 1997).

Nicolls et al. (2006) investigated "post-midnight uplifts" using ISR and ionosonde data from Brazil and Peru, while Chau et al. (2009) focused on vertical ExB drifts during a sudden stratospheric warming event, highlighting the ISR's capacity to capture short-term events.

# 1.3 Digisonde and Ionogram-Based Measurements

Another technique for measuring vertical drift uses Digisonde Drift Measurements (DDM). Unlike traditional ionosondes that record only the time-of-flight of reflected signals (ionograms), Digisondes can detect specific reflection points and measure Doppler shifts at those points, enabling full vector drift estimations. However, for vertical drift in particular, this direct measurement does not distinguish whether the observed Doppler shift is caused by the actual motion of ionospheric plasma or by changes in the reflection height due to ionization and recombination processes. This effect must be considered carefully when interpreting vertical drift data. Digisonde drift measurements are performed in a dedicated fixed-frequency mode that does not produce ionograms during the measurement intervals and therefore do not provide information about the full ionospheric profile.

Despite this limitation, DDM data are especially valuable for tracking TIDs. For instance, Altadill et al. (2007) analyzed 18 months of DDM data from the Ebro Observatory, revealing significant seasonal differences in drift patterns. Kouba et al. (2016) conducted a study at the mid-latitude Pruhonice station, identifying characteristic daily variations, such as a pronounced early-morning negative peak and a gradual positive shift toward local noon. In addition, recent work by Ma et al. (2022) introduced a fully automated data processing method for drift measurements, enabling robust extraction of drift velocity vectors and reducing the need for manual parameter tuning.

Even classical ionograms, lacking Doppler and reflection point information, can still be used to estimate vertical drift. Researchers such as Abdu et al. (2004), Mathew et al. (2010), and Kelley (2009) used temporal changes in virtual heights (e.g., h'F) as a proxy for vertical drift. These studies revealed both seasonal and solar activity-related variability in vertical motion.

# 1.4 Comparative Analyses and Motivation of This Study

Several studies have attempted to compare different approaches to evaluate their consistency. Woodman et al. (2006) compared vertical and zonal drift measurements obtained from the Jicamarca ISR with those derived from the Digisonde, focusing on a few case-study days. Their results showed good agreement for the vertical drifts at periods when convection dominates (e.g., during the nighttime and the pre-reversal enhancement), whereas the daytime correspondence when

production and recombination are not dominant was generally poor. Yue et al. (2008) analyzed the correspondence between Fejer–Scherliess empirical model predictions of the equatorial E×B drift and hmF2-derived variations, finding that the model and hmF2-based drifts agreed well during specific local-time intervals (around sunrise and sunset). These comparisons illustrate that while different methods may not always agree in detail, they often preserve key temporal trends and offer complementary perspectives on ionospheric dynamics.

It is generally difficult to validate results across different techniques because suitable data are rarely available simultaneously. In this study, we aim to address this gap by conducting a detailed comparison of vertical drift estimates obtained from ionosonde data (derived from ionogram analysis) and direct drift measurements provided by the Digisonde system.

Since the Digisonde is a digital ionosonde with additional drift-measurement capability (DDM), it offers an ideal platform for comparing the results of both approaches using data from the same instrument. These data will then be compared across different temporal scales, enhancing our understanding of both method-specific uncertainties and the underlying plasma dynamics.

## 2 Methods

100

105

120

125

## 2.1 Digisonde Drift Measurement (DDM)

The Digisonde technique, a specialized ground-based measurement method, plays a pivotal role in determining plasma drift velocities. It involves a meticulous analysis of signals reflected from the ionosphere at a selected sounding frequency. This analysis precisely pinpoints the location of the reflection points within the ionosphere, along with the corresponding Doppler shift values. Such a data set allows accurate estimation of the drift velocity vector, a technique originally developed by Wright and Pitteway (1994) and later expanded by Reinisch et al., 1998. This process stands as a crucial step in the automatic data processing of DDM (Kozlov and Paznukhov, 2008).

The locations of the reflection points are visually represented in a graphical display known as SKYmap.

The Doppler frequency shifts  $(D_i)$  for individual echoes are proportional to the line-of-sight velocity  $(D_i=-2f_0/c(\mathbf{k_i}\cdot\mathbf{v}))$ , where  $f_0$  is the sounding frequency, c is the speed of light,  $\mathbf{v}$  is the drift velocity vector, and  $\mathbf{k_i}$  is the directional vector corresponding to the i-th individual echo. Since a large number of individual reflection points are typically detected during each measurement, the drift vector estimation represents an overdetermined problem. The velocity vector  $\mathbf{v}$  is usually derived using the least squares method from all detected reflection points (DDA method) (Reinisch et al., 1998, Reinisch et al., 2005). The typical outcome of DDM measurements is a drift velocity vector containing the vertical component  $(v_z)$  and two horizontal components  $(v_N, v_E)$ , or alternatively, the magnitude and azimuth of the horizontal component. In such a framework, it is also possible to estimate the uncertainty of each velocity component.

One approach to assess this uncertainty is through resampling techniques, such as bootstrap (Efron and Tibshirani, 1993) or jackknife (Efron, 1982) methods. In our case, we used repeated subsampling of the detected reflection points — that is, selecting multiple random subsets from the original set of echoes and recomputing the velocity vector for each subset. Statistical analysis of these velocity estimates (e.g., computing their standard deviation) then provides an empirical estimate of the measurement uncertainty for each velocity component.

The accuracy of DDM measurements is influenced by several factors. First and foremost, the appropriate choice of sounding frequency  $f_0$  is critical. The autodrift mode is commonly used for measurements in the F region, where the sounding frequency is automatically determined based on the critical frequency of the F layer (foF2) obtained through the autoscaling process from the latest measured ionogram. However, difficulties may arise if the autoscaling process fails or the critical frequency is estimated incorrectly.

The second crucial factor affecting the acquisition of the drift velocity vector is the number and spatial distribution of reflection points detected during DDM (character of SKYmap). Successful determination of all components of the drift velocity vector with small error is associated with the detection of SKYmaps with a large number of reflection points distributed over a wide spatial area. Such SKYmaps are typically obtained in a disturbed ionosphere, particularly during spread-F conditions. In contrast, in quiet ionospheric conditions, the reflected signals tend to be concentrated in a near-vertical direction, which limits the spatial distribution of reflection points. As a result, SKYmaps in these conditions often display only a small cluster of reflection points near the vertical direction. In this case, determining horizontal components of the drift velocity becomes practically impossible or results in significant errors. However, the vertical component can be still determined with sufficient accuracy (Kouba and Koucká Knížová, 2012).

The third factor that can prevent the measurement of the vertical drift velocity component is the presence of a sporadic E layer, whose total blanketing can completely inhibit measurements. Additionally, when multiple Es (or other multiples) are close to the sounding frequency, it becomes challenging or even impossible to determine the correct reflection. Automatic calculations may not reveal this issue, requiring manual evaluation for accurate results. For further processing, refer to Kouba et al. (2008).

As previously mentioned, the vertical component of the velocity obtained in this manner does not necessarily correspond solely to the motion of the plasma. From this perspective, the term 'drift velocity' may not be entirely appropriate. In cases where the observed Doppler shift is induced by factors such as variations in ionization, the term 'apparent drift velocity' is sometimes used (Scali and Reinisch, 1995; Mridula and Pant, 2022). In our study, we compare vertical velocity values obtained through multiple methods without interpreting their physical origin. This approach is focused on comparing the different methods used to estimate vertical drift, which is why distinguishing between actual plasma motion and apparent drift velocity is not necessary for our analysis. Therefore, in the following text, we use the term 'vertical drift' without distinguishing between these two contributions.

# 2.2 Indirect ionogram-based methods








These methods rely on the time variation of certain characteristic heights, denoted as  $h_X$ . For consecutive ionograms with a time difference  $\Delta t$ , a specific characteristic height for the F-layer of the ionosphere,  $h_X$ , is determined. The vertical drift velocity component  $(v_z)$  can then be obtained as  $v_z = \Delta h_X/\Delta t$ .

Various authors (Prabhakaran Nayar et al. (2009), Oyekola and Kolawole (2010), Adeniyi et al. (2014), Adebesin et al. (2015), Simi et al. (2014), Saranya et al. (2014), Bertoni et al. (2011) for instance) choose different variables for h<sub>x</sub>, such as hmF2, h'F2, h'(3.5 MHz), h'(0.8\*foF2), etc., in their research papers. Some of these characteristic heights are depicted/highlighted on specific ionogram/electron concentration profile in Figure 1.

One commonly used characteristic height is the true-height (hmF2) of the F2 peak on the electron density profile. However, the real-height electron density profile is not a direct outcome of vertical ionospheric sounding but it is inverted from the ordinary trace recorded on the ionogram. The computation of hmF2 values is usually carried out based on the knowledge of the complete ordinary mode trace in the ionogram (Reinisch and Xueqin,1983; Reinisch andHuang, 2001). The precise value of hmF2 depends on the quality and methodology of the trace scaling, particularly near the critical frequency - that is, up to the virtual height to which the trace is extended. A comparison of results obtained using the POLAN (POLynomial ANalysis) and ARTIST (Automatic Real-Time Ionogram Scaler with True Height) algorithms is presented in Šauli et al. (2007). Further details on POLAN can be found in Titheridge (1985), and on ARTIST in Reinisch (1996) and Reinisch et al. (2005), among other sources.

An advantage of hmF2 as a parameter is that it is directly related to the maximum electron concentration - a physically well-defined quantity. On the contrary however, the ionospheric F layer is variable, therefore changes in the maximum height during the day or even between two consequent measurements should be considered.







Another frequently used parameter is the minimum virtual height of the F2-layer (h'F2) ordinary wave trace on the ionogram (Piggott and Rawer, 1972). This height is one of the fundamental characteristics determined on the ionogram since the beginning of regular ionospheric measurements in the 1930s. Currently, the autoscaling of h'F2 generally works effectively.

The advantage of choosing the h'F2 is its easy accessibility, and availability of a long time series. Values of h'F2 from numerous world stations are stored in ionospheric databases, such as the Global Ionosphere Radio Observatory (GIRO) [Reinisch and Galkin, 2011] and the Digital Ionosonde Database (DIAS) [Belehaki et al., 2005].

The parameter h'(f) represents the virtual heights of the F2 layer recorded on ionograms for a selected fixed sounding frequency f. In the presented comparison, the sounding frequency of 3.5 MHz is used. The sounding frequency of 3.5 MHz is one of frequencies used by Prabhakaran Nayar et al. (2009). The advantage of this simple method is that it does not require classical ionospheric sounding (complete ionogram measurement requiring a band of sounding frequencies). For the selected sounding frequency only the time of arrival of the reflected signal reflected from the ionosphere needs to be measured. Thus, it is suitable for campaigns conducted in locations without ionosondes, where measurements can be made with simpler equipment. However, a significant drawback is the uncertainty in determining which part of the trace on the ionogram is being measured, and thus, identifying the corresponding reflection region of the ionosphere. The obtained height may differ significantly from the height of the maximum electron concentration. Another limitation of this approach is, that without knowledge of the complete ionogram, it is even impossible to determine if the measurement is within the F1 trace, F2 trace, multiple Es trace or even regular E layer.

Further, parameter h'(0.8foF2) represents the virtual height of the F2 layer measured on the ionogram for the frequency 0.8foF2. Unlike h'F2, the value is not widely available. First, the value of the critical frequency foF2 needs to be determined, and then the virtual height of the ordinary mode trace for the frequency 0.8foF2 is extracted from the ionogram. An advantage of this method could be that it surpasses some of the limitations of the above-mentioned parameter h(3.5 MHz). Measurement is performed relatively close to the maximum electron concentration, while avoiding some of the issues related to where exactly the trace scaling ends (a problem with hmF2) and reflection from Es or regular E layer.

Fundamentally, an ionogram provides the virtual height at which a signal is reflected for a given sounding frequency. When analyzing the temporal variation of this height, it becomes evident that, without additional information, it is not possible to

determine whether the observed changes result from the vertical motion of the plasma or other processes. Therefore, the consideration regarding apparent drift, as discussed earlier, is also relevant in this context.

#### 205 3 Measurements






# 3.1 High sampling rate case studies

In this study we focus on a period of a rather quiet ionosphere, when we could concentrate on principal properties of the compared methods. Under such conditions we may expect to obtain good quality ionograms and drift data. It means, the study is not contaminated by unusual stratification of the electron concentration with well defined reflection planes for sounding radio waves.

For our study, we utilized data from two one-day special high-rate campaigns conducted at the ionospheric observatory in Průhonice (a mid-latitude station in the European sector with geographic coordinates: latitude =  $N50.0^{\circ}$ , longitude =  $E14.6^{\circ}$  - geomagnetic coordinates: latitude =  $49.553^{\circ}$ , longitude =  $98.236^{\circ}$ ).

During these campaigns, both high-temporal-resolution vertical ionospheric sounding (with intervals of 1 or 2 minutes respectively) followed by direct drift measurements for the F region were performed. Particular setting of DPS 4D is provided in Table 1.

The first campaign (Campaign I) took place on October 24, 2017 (day 297), during the declining phase of the solar cycle 24 (SC24). The geomagnetic activity during the campaign was low to moderate, with the Kp index ranging from 1- to 5, and the Ap index reaching a value of -39. The second selected campaign (Campaign II) was performed on March 20, 2023 (day 79), near the maximum of the solar cycle 25 (SC25). The geomagnetic activity, characterized by the Kp index, ranged from 0+ to 4, and the Ap index reached a value of -27.

We emphasize that this study is based on two targeted one-day campaigns, selected to provide high-cadence, high-quality measurements under relatively quiet ionospheric conditions. While the dataset is limited in scope, it allows detailed side-by-side evaluation of methods at minute-scale resolution. We do not aim to generalize the findings but rather to identify specific methodological discrepancies that warrant further investigation with broader datasets.

# 3.2 Campaign setting description

Historically, standard ionogram soundings were typically performed at 15-minute intervals at most ionospheric stations worldwide. Hourly values of key ionospheric parameters were then usually manually scaled and submitted to central databases. When using a Digisonde in standard operating mode, a single ionogram measurement typically takes about 1–2 minutes, depending on the specific configuration (such as the range of sounding frequencies and the frequency step). In recent years, some stations have adopted denser sounding schedules, with standard cadences of 5 minutes. During the Campaign I, ionograms and drifts in the F region were measured at a cadence of 1 minute. To enable such short duration measurements, particular non-standard measurement settings were employed: only the ordinary signal was recorded on the ionograms, in order to reduce the measurement time by half compared to the usual practice of detecting both ordinary and extraordinary polarizations. Further, a less fine frequency step for precise height measurements was used, primarily to achieve measurement in a time window shorter than half a minute. For the drift measurements in the F region with a

sounding cadence of 1 minute, the measurement time needed to be restricted to half a minute. As a result, the drift measurements were substantially modified compared to normal conditions. Both the number of measurement repetitions and frequency steps were limited. During the nighttime schedule, drifts were measured in a fixed frequency range of approximately 1.8 - 4 MHz, while during the daytime schedule, they were measured in a range of 3 - 6 MHz.

During Campaign II, ionograms were measured at a cadence of 2 minutes, with an additional series of three drift measurements in the F region conducted between each pair of ionograms. In this case, ionograms were measured for both polarizations - the ordinary and extraordinary modes. The used setting allows to determine the height more accurately as a fine frequency step was utilized. During the nighttime schedule, drift measurements were performed in autodrift mode and in two narrow frequency windows around 3.5 and 4.5 MHz. In the daytime schedule, drift measurements were conducted in autodrift mode and in two narrow frequency windows around 4.5 and 7 MHz.

The specific settings for the ionogram measurements are listed in Table 1.

#### 4 Data






# 4.1 Vertical drift vz as a result of DDM

In both campaigns the time schedule was modified to satisfy high temporal resolution compared to regular sounding. Therefore, the duration of each individual drift measurement was reduced to achieve higher temporal resolution. As a result of these limitations, a smaller number of reflection points were detected, leading to lower quality drift data in general. Typically, only a few tens of reflection points were detected during a single measurement, and under such circumstances, it is not feasible to determine the drift velocity vector with high precision. A smaller number of detected reflection points can significantly influence the accuracy of the drift velocity calculations, as the precision of the measurements depends heavily on the spatial coverage. For that, longer measurements and favorable conditions for detecting points over a wider area would be required (Kouba and Koucká Knížová, 2012). However, for our study, which solely utilizes the vertical component of the velocity, the employed measurements are entirely adequate. The small number of detected reflection points with insufficient spatial coverage primarily manifests in the reduced accuracy of calculating the horizontal components of the drift velocity. In the vast majority of cases, the accuracy of determining the vertical component is sufficient (Kouba and Koucká Knížová, 2012).

In both campaigns, the vertical component of the drift velocity was computed for each successful DDM measurement. Measurement failures were only observed in exceptional cases, such as auto-scaling errors that led to incorrect determination of the sounding frequency. Consequently, a time series of one minute values for the vertical component of the drift velocity for the first campaign, and a time series of three values for the vertical component of the drift velocity within two minutes for the second campaign were obtained.

These time series provide a valuable foundation for further examining the temporal variations and dynamics of the vertical drift component.

Figure 2 presents the results of direct vertical drift measurements (DDM) obtained on 24 October 2017. The top panel shows the unsmoothed data, while the subsequent plots display data smoothed using moving averages over 5, 15, 30, and 60

minutes. Across all panels, the general temporal pattern of the vertical plasma drift remains preserved; longer smoothing intervals progressively reduce short-term fluctuations and high-frequency noise, while retaining the larger-scale structures. Each data point is accompanied by a vertical error bar representing the standard deviation of the individual detections within each measurement window. These error bars reflect the internal variability of the detected reflection points at a given time. Notably, in some parts of the day, the uncertainties are significantly larger—especially in the raw (unsmoothed) data—while in other intervals they remain relatively small.

This variability is primarily related to the number of detected echo points during each measurement. A low number of reflections typically leads to large uncertainties due to reduced statistical confidence in the vertical drift estimate. In contrast, when many reflections are detected, the derived drift value is more stable and the standard deviation correspondingly smaller.

It is important to note that during this special measurement campaign, a very short measurements was used for both ionograms and drift measurements, resulting in reduced sounding time and fewer detected points. In contrast, during regular routine operation, each measurement is based on a significantly longer sounding sequence. Therefore, under standard measurement conditions—typically with repetition intervals of 5 to 15 minutes—substantially lower uncertainties can be expected, leading to more precise drift estimates.

#### 290 4.2 Vertical drift component v<sub>z</sub> obtained using ionogram characteristics




Each ionogram obtained during both campaigns was manually scaled. The ordinary trace was processed using the ARTIST algorithm within the SAO Explorer (Khmyrov et al., 2008), yielding the electron concentration profile.

From each ionogram or its corresponding electron density profile, characteristic height parameters were extracted and subsequently used to compute the vertical drift velocity using various indirect methods. These parameters include hmF2, which represents the height of the maximum electron concentration, h'F2, the virtual height of the F2 layer, h'(3.5 MHz), the virtual height of the ionogram trace for the sounding frequency of 3.5 MHz, and h'(0.8foF2), the virtual height corresponding to 0.8 times the critical frequency foF2. The subsequent analysis is described in detail for the case of hmF2; however, the same processing approach was applied to the time series of all other height-related quantities.

By using hmF2 values from two ionograms measured at different times, the apparent vertical drift velocity ( $v_z$ ) was calculated according to the formula  $v_z$ = $\Delta$ hmF2/ $\Delta$ t. Applying this relation to pairs of consecutive ionograms yielded a time series of apparent vertical drift velocity with one-minute resolution for Campaign I and two-minute resolution for Campaign II. It should be emphasized that this approach provides an apparent drift, reflecting the temporal change of the F2-layer peak height rather than the true plasma motion. As discussed in Sect. 1.4 and by Bittencourt and Abdu (1981), such ionosondederived F-region drifts may not represent actual plasma motion during periods of dominant production and recombination (typically in local daytime). Therefore, interpretation of these results must consider these limitations when comparing with direct drift measurements. The resulting time series for Campaign I is shown in Figure 3A. In this figure, an unrealistic temporal evolution of the  $v_z$  component is clearly visible, characterized by large variations lacking any coherent temporal structure. The unrealistic temporal evolution refers to large fluctuations in the drift velocity that do not correspond to the expected physical behavior of the ionosphere, suggesting that errors in the measurement intervals contribute

- disproportionately to these variations. It is evident that this representation does not reflect the true development of vertical drift velocity. The primary reason lies in the short time intervals between the ionograms used for the calculation. The determination of individual ionogram parameters is inherently imprecise; in particular, it is unrealistic to assume an uncertainty in hmF2 of less than 1 km. In practice, the uncertainty is often larger due to several factors, including ambiguous identification of the F2 trace and the sensitivity of the ARTIST algorithm to other ionospheric features.
- Although the uncertainty in the determination of the characteristic height cannot be significantly reduced, the resulting error in velocity estimation can be mitigated by appropriately selecting the time interval Δt. When the interval between two ionograms is too short—on the order of tens to a few hundreds of seconds—the relative contribution of the height uncertainty becomes dominant, leading to substantial errors in the computed drift velocity. Therefore, longer time intervals are preferable for obtaining more reliable velocity estimates from indirect methods.
- In practice, the goal is to analyze the obtained time series and detect irregularities, wave structures, and other dynamic features in the data. Therefore, it is essential to find an optimal time step that ensures reliable precision in the measurements, making the time series consistent. However, the time step should not be so large that important short-term details, which are crucial for detecting these irregularities, are lost. A time step that is too large would smooth out short-term oscillations, which are crucial for detecting wave phenomena, while a too-small interval can lead to noisy data that is difficult to interpret.

  Balancing the need for accuracy with the preservation of fine temporal details is critical for effective analysis.
  - To illustrate the impact of time step selection on the resulting time series, Figure 3A–E presents the derived vertical drift velocity  $v_z$  obtained with different temporal resolutions. Specifically, panel A shows a time series constructed from ionograms with a one-minute interval, panel B with a five-minute interval, and panels C to E with intervals of 15, 30, and 60 minutes,
- 330 It is clearly visible that the characteristics of the time series change significantly with increasing time steps. Even with a five-minute interval (Fig. 3B), the values exhibit considerable variability, although some segments already indicate a systematic temporal evolution. At a 15-minute interval (Fig. 3C), this systematic behavior becomes more pronounced and persists with the longer intervals of 30 and 60 minutes (Figs. 3D, E), indicating that these resolutions are sufficient for capturing the underlying trends.
- Importantly, the 15-minute interval provides a suitable balance between preserving dynamic features and minimizing random noise. Therefore, in the subsequent analysis, we focus on time series obtained from 15-minute sampling, further smoothed with a matching 15-minute moving average to suppress short-term fluctuations and emphasize meaningful structures.
- This approach is particularly justified by the nature of the data processing itself: when using a 15-minute interval between two ionograms (i.e., using values derived from measurements at times t and t + 15 min), the resulting drift value inherently reflects an average over that time span. On the other hand, if we work with higher temporal resolution, such as one ionogram per minute, we obtain 15 individual drift estimates within each 15-minute window. However, since these values are derived

from overlapping measurement pairs within the same window, they do not provide independent information on sub-window scale variability.

Consequently, applying a moving average with the same 15-minute window serves not only to suppress random variability but also to reinforce consistency between the inherent resolution of the estimates and the desired smoothing.

For clarity, we note that the vertical drift values were first computed as differences of height parameters (e.g.,  $\Delta hmF2$ ) measured at time-separated ionograms, using predefined time intervals  $\Delta t$  (1, 5, 15, 30, or 60 minutes). The resulting time series was then smoothed using a centered moving average. We also tested the reverse procedure—smoothing the height parameters first and then computing drift—and found no significant differences in the final smoothed time series. We therefore retained the direct-differencing approach for its simplicity and better control over temporal structure.

Figure 4 illustrates the effect of different smoothing windows applied to the hmF2 parameter, using time windows of 1 minute (Fig. 4A), 5 minutes (Fig. 4B), and longer intervals. It is evident that short smoothing intervals (1 and 5 minutes) do not sufficiently suppress noise and fail to produce consistent trends. In contrast, smoothing over 15 minutes or longer enhances the temporal coherence of the data.

# 5 Results







The comparison of time series obtained from different methods reveals significant differences in reliability and consistency, particularly at short timescales. A key observation is the high temporal coherence and stability of the results derived from DDM, which remain consistent across various temporal resolutions. In contrast, indirect methods based on ionogram-derived parameters — such as hmF2, h'F2, or h' at fixed frequencies — exhibit substantial variability when applied to short sampling intervals. As previously discussed, this variability leads to significant errors that compromise their ability to resolve short-period oscillations, rendering the results unreliable on timescales shorter than approximately 15 minutes.

Based on these findings, we adopt a 15-minute sampling interval in the subsequent analysis, combined with an additional 15-minute moving average. This dual-step approach is justified and not redundant: while the sampling interval ensures each point integrates over sufficient data to suppress random fluctuations, the moving average further reduces residual short-term variability within each window. Together, they enhance the visibility of persistent and physically meaningful structures in the data.

Figures 5 and 6 show results for Campaigns I and II, respectively. In both cases, the upper panels display raw time series (without smoothing), while the lower panels present the same data after applying a 15-minute moving average (16-minute for Campaign II). The black dots represent the vertical drift velocity component  $(v_z)$  from DDM. The light blue, red, violet, and green lines correspond to  $v_z$  derived from hmF2, h'F2, h'(3.5 MHz), and h'(0.8 foF2), respectively.

As illustrated in Figure 5, the largest discrepancies between methods — in both amplitude and trend — occur mainly during daytime hours, particularly from 04 to 15 UT. In contrast, during nighttime periods (00–04 UT and 15–24 UT), the agreement between the methods improves considerably. This is especially apparent in the smoothed time series (bottom panels), where the general trend of the vertical drift velocity  $v_z$  is more consistent across all methods. In the unsmoothed data (top panels), differences in both high-frequency variability and trend direction are more pronounced.

One notable outlier is the purple curve in Figure 5, representing the h'(3.5 MHz) method. This method exhibits frequent and significant deviations from the others, especially during nighttime hours. This can be attributed to the proximity of the sounding frequency (3.5 MHz) to the critical frequency foF2. For example, between 00 and 05 UT, foF2 ranges from 2.7 to 3.8 MHz, while after 20:30 UT it again approaches 3.5 MHz. When the sounding frequency approaches foF2, the reliability of the inferred virtual height — and consequently the derived vertical drift — is compromised.

By contrast, the results obtained during Campaign II (Figure 6) show improved agreement across all methods, including h'(3.5 MHz). This is likely due to the fact that foF2 remained consistently above 4.3 MHz throughout the analyzed interval. As a result, the 3.5 MHz frequency was sufficiently below foF2 to allow for more accurate height estimation, leading to improved consistency in derived drifts.

The general patterns observed in Campaign I are confirmed in Campaign II: large discrepancies between methods are again observed during daytime hours, particularly between 06 and 11 UT, while better consistency appears during the night (00 - 04 UT) and after 15 UT. Notably, in the smoothed series, both the amplitude and shape of the drift curves align more closely across all methods, confirming the utility of the 15–16 minute moving average in suppressing short-term noise and enhancing the detection of physically relevant features.

# **6 Discussion**






The comparison of vertical drift estimates obtained from direct and indirect methods reveals both the limitations and complementary value of ionogram-based techniques. While DDM provides stable and consistent results across all time scales, indirect methods relying on characteristic ionospheric heights (e.g., hmF2 or h' at fixed frequencies) are susceptible to significant errors at short temporal resolutions due to their sensitivity to signal quality and the proximity of the sounding frequency to foF2, which may result not only in quantitative uncertainties but also in misinterpretation of apparent height variations as true vertical plasma motion.

Our findings indicate that indirect methods, when applied to high-cadence data, often produce inconsistent and physically implausible drift patterns. These inconsistencies are especially evident in the presence of noise-induced artifacts that resemble wave-like structures but lack consistency across different estimation techniques. Such artifacts can lead to misleading scientific interpretations if not carefully examined. Among the tested techniques, the method using h(3.5 MHz) proved particularly unreliable when foF2 approached 3.5 MHz, a situation in which the method consistently failed to provide meaningful results. In this regard, the hmF2-based approach appears to be the most stable among the indirect techniques.

The use of indirect methods becomes practically infeasible at very short time steps. Our findings suggest that a temporal resolution of approximately 15 minutes represents the practical lower limit for obtaining consistent results with indirect methods. In contrast, a 5-minute step—which corresponds to the standard ionogram cadence at some stations—still frequently yields unstable and inconsistent drift estimates. Therefore, in the following discussion we focus on results derived from 15-minute averaged inputs. Even at this time scale, however, notable discrepancies between the individual methods persist in many cases.

Nevertheless, during certain periods—particularly at night—all methods exhibit very good agreement in both amplitude and trend. This consistency across different sounding frequencies, and thus across different altitudes within the F2 layer, indicates coherent vertical drift behavior throughout the layer. In such cases, it is reasonable to interpret the derived values as representing the actual vertical plasma velocity, especially when nighttime conditions above 300 km are met (Bittencourt and Abdu, 1981). This fact provides excellent opportunities to expand the dataset available for various regional and global models that rely on accurate vertical drift inputs.





In contrast, typically during daytime hours, pronounced discrepancies between methods are frequently observed—both in the magnitude and temporal evolution of the vertical drift velocity  $v_Z$ . This suggests the presence of distinct physical processes occurring at different altitudes. Under these conditions, the obtained values should not be interpreted as direct measurements of vertical plasma motion. Essentially, ionosonde-derived drifts based on successive heights tend to approach zero during local daytime, as the ionospheric electron density profile may remain close to equilibrium despite the presence of actual plasma motion. Consequently, the apparent change in virtual or true height between consecutive ionograms becomes very small, leading to unrealistically low drift values. This limitation, already discussed by Bittencourt and Abdu (1981) and further demonstrated by Woodman et al. (2006), emphasizes that ionosonde-derived F-region drifts may not reflect the true plasma motion under strong production and recombination conditions. Instead, they often reflect apparent drifts, which result from shifts in the virtual reflection height caused by local ionization and recombination processes. Although these apparent drifts do not represent true plasma motion, they remain highly valuable: their analysis enables the detection and tracking of wave-like structures in different regions of the ionosphere.

As clearly demonstrated by the presented data, wave activity is frequently observed throughout the analyzed intervals. The various methods used respond differently to specific processes depending on their sensitivity to different ionospheric altitudes. Despite these differences, dominant wave patterns are consistently captured by all techniques, offering a robust multi-method approach for identifying, tracing, and characterizing ionospheric wave phenomena. The application of a 15–minute (16-minute) moving average further enhances this capability by suppressing high-frequency noise and emphasizing persistent, physically meaningful features.

While our analysis is based on only two one-day campaigns, the combination of high-cadence DDM and ionogram observations under quiet conditions offers a uniquely controlled tested. This focused setup allows us to isolate method-inherent discrepancies and assess the stability of derived signatures in a consistent observational environment. The aim of this study was not to provide an exhaustive validation against all available techniques, but rather to highlight the intrinsic behavior, limitations, and potential inconsistencies of commonly used ionogram-based drift estimation methods. A detailed validation against incoherent scatter radar (ISR) data, would certainly be valuable for future work and would help to further quantify the reliability of different methods across varying geophysical conditions. Such targeted case studies form a necessary first step toward establishing reliable validation strategies for vertical drift estimation methods using more extensive and diverse datasets in the future.

- 445 Acknowledgments. DK acknowledges the support of the ESA Project SWESNET (4000134036/21/D/MRP). The work of PKK was partly supported by the ESA project QUID-REGIS ESA Contract No. 4000143632/24/I-EB. ZM expresses gratitude for the project CZ.02.01.01/00/22\_008/0004605 financed by the Ministry of Education, Youth, and Sports.
- We would also like to express our gratitude to the Ionospheric Observatory in Pruhonice for providing the ionospheric data used in this study, accessible through the GIRO database (<a href="https://hpde.io/SMWG/Observatory/GIRO">https://hpde.io/SMWG/Observatory/GIRO</a>).

Furthermore, we extend our appreciation for the valuable geomagnetic indices Kp and Dst used in this research. The source of these indices was obtained from the Kyoto database (WDC for Geomagnetism, Kyoto (kyoto-u.ac.jp)).

Author contributions. DK conceptualized the study, conducted the experiments, and wrote the first draft of the manuscript.
 Data analysis was performed by DK and ZM. PKK and ZM contributed to the revisions of the manuscript.

Competing interests. The authors declare no competing interests.

| ionogram<br>measurements<br>settings | 24.10. 2017<br>Campaign I |           | 20.3. 2023<br>Campaign II |           |
|--------------------------------------|---------------------------|-----------|---------------------------|-----------|
|                                      | daytime (6-17 UT)         | nighttime | daytime (7.30-18.30 UT)   | nighttime |
| starting frequency                   | 1 MHz                     | 0.5 MHz   | 1 MHz                     | 1 MHz     |
| ending frequency                     | 9 MHz                     | 5 MHz     | 14 MHz                    | 7.5 MHz   |
| Frequency step                       | 0.05 MHz                  | 0.025 MHz | 0.1 MHz                   | 0.05 MHz  |
| Fine frequency step                  | Х                         | X         | 5 kHz                     | 5 kHz     |
| Polarisations                        | О                         | О         | O,X                       | O,X       |

Table 1. Settings of key ionogram measurement parameters for the high-rate campaigns on October 24, 2017, and March 20, 2023.

Figure 1: In the ionogram recorded in Pruhonice on March 17, 2017, and on the corresponding electron concentration profile, the parameters peak height F2-layer (hmF2), minimum virtual height of F2 trace (h'F2), virtual height of plasma with plasma frequency 3.5 MHz h'(3.5MHz), and virtual height of plasma with 80 % of critical frequency foF2 (h'(0.8\*foF2)) are depicted.

Figure 2: Vertical drifts measured by Digisonde (DDM) during the Campaign I, successively: all unsmoothed measurements, smoothed with a 5-minute window, 15-minute window, 30-minute window, and 60-minute window. Each data point includes error bars representing measurement uncertainty.

Figure 3: Vertical drifts computed using the hmF2 parameter for the minute-sampling Campaign I. Calculations were performed with measurement intervals of 1 minute (panel A), 5 minutes (panel B), 15 minutes (panel C), 30 minutes (panel D), and 60 minutes (panel E) without any smoothing.

Figure 4:Calculation of vertical drifts derived using the hmF2 parameter for the minute-sampling Campaign I. The calculation was conducted with measurement intervals of 1 minute, 5 minutes, 15 minutes, 30 minutes, and 60 minutes, with the employed smoothing corresponding to each respective time step.

Figure 5: Comparison of vertical drifts obtained for Campaign I using various methods with a 15-minute measurement interval:

485 DDM - black, green - hmF2, blue - h'F2, purple - h'(3.5 MHz), orange - h'(0.8foF2). No smoothing is applied in the upper panel, while a 15-minute smoothing window is used in the lower panel.

490 Figure 6: Comparison of vertical drifts obtained for the Campaign II using various methods with a 16-minute measurement interval, indicated similarly to Figure 5.

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
