# Peer review of "Apparent Vertical Ionospheric Drift: A Comparative Assessment of Digisonde and Ionogram-Based Methods"

_EGUsphere, 2025_

## Author Response (AR1)

**Response to Referees**

We thank both reviewers for their detailed evaluations and constructive feedback. Below we provide a revised point-by-point response that accurately reflects the changes made in the manuscript, as confirmed by the submitted PDF with tracked changes. All colored figures were recolored to follow colorblind-friendly palettes and time series plots were adjusted for improved readability.

**Reviewer 1**

**Major Comment 1:**

Reviewer: The study relies on only two days of observational data...

**Response:** We agree with the reviewer that the limited data volume constrains the generalizability of our conclusions. However, our aim was not to provide an exhaustive intercomparison across all geophysical conditions, but rather to highlight the limitations and discrepancies among methods under quiet, controlled conditions. This intent is explicitly discussed in the manuscript.

**Changes in Manuscript:**

• Lines 435–439 (line numbers always related to pdf with change-tracking) (Section 5): Added text: "While our analysis is based on only two one-day campaigns, the combination of high-cadence DDM and ionogram observations under quiet conditions offers a uniquely controlled tested. This focused setup allows us to isolate method-inherent discrepancies and assess the stability of derived signatures in a consistent observational environment. Such targeted case studies form a necessary first step toward establishing reliable validation strategies for vertical drift estimation methods using more extensive and diverse datasets in the future."

**Major Comment 2:**

Reviewer: Include DDM error bars in Figures 5 and 6.

**Response:** We have included vertical error bars representing standard deviation estimates for the DDM-derived vertical drifts. These values were computed from the residuals of the least-squares fitting process.

**Changes in Manuscript:**

- Figures 5 and 6 updated to include DDM error bars.
- Line 270-282 (Section 4.1): Added: "Each data point is accompanied by a vertical error bar representing the standard deviation of the individual detections within each measurement window. These error bars reflect the internal variability of the detected reflection points at a given time. Notably, in some parts of the day, the uncertainties are significantly larger—especially in the raw (unsmoothed) data—while in other intervals they remain relatively small.

This variability is primarily related to the number of detected echo points during each measurement. A low number of reflections typically leads to large uncertainties due to reduced statistical confidence in the vertical drift estimate. In contrast, when many reflections are detected, the derived drift value is more stable and the standard deviation correspondingly smaller.

It is important to note that during this special measurement campaign, a very short measurements was used for both ionograms and drift measurements, resulting in reduced sounding time and fewer detected points. In contrast, during regular routine operation, each measurement is based on a significantly longer sounding sequence. Therefore, under standard measurement conditions—typically with repetition intervals of 5 to 15 minutes—substantially lower uncertainties can be expected, leading to more precise drift estimates."

**Major Comment 3:**

Reviewer: Authors avoid strong critical commentary on performance of the tested methods.

**Response:** We agree and have revised the Discussion section to clearly articulate the practical limitations of each method, particularly under high-resolution settings. Recommendations are added for choosing appropriate methods under different conditions.

**Changes in Manuscript:**

• Lines 396-415 (Section 6): Rewrited and added: "While DDM provides stable and consistent results across all time scales, indirect methods relying on characteristic ionospheric heights (e.g., hmF2 or h' at fixed frequencies) are susceptible to significant errors at short temporal resolutions due to their sensitivity to signal quality and the proximity of the sounding frequency to foF2, which may result not only in quantitative uncertainties but also in misinterpretation of apparent height variations as true vertical plasma motion.

Our findings indicate that indirect methods, when applied to high-cadence data, often produce inconsistent and physically implausible drift patterns. These inconsistencies are especially evident in the presence of noise-induced artifacts that resemble wave-like structures but lack consistency across different estimation techniques. Such artifacts can lead to misleading scientific interpretations if not carefully examined. Among the tested techniques, the method using h(3.5 MHz) proved particularly unreliable when foF2 approached 3.5 MHz, a situation in which the method consistently failed to provide meaningful results. In this regard, the hmF2-based approach appears to be the most stable among the indirect techniques.

The use of indirect methods becomes practically infeasible at very short time steps. Our findings suggest that a temporal resolution of approximately 15 minutes represents the practical lower limit for obtaining consistent results with indirect methods. In contrast, a 5-minute step—which corresponds to the standard ionogram cadence at some stations—still frequently yields unstable and inconsistent drift estimates. Therefore, in the following discussion we focus on results derived from 15-minute averaged inputs. Even at this time scale, however, notable discrepancies between the individual methods persist in many cases."

**Minor Comments:**

L. 73–75: Clarify fixed-frequency requirement for Digisonde drift mode.

Response: Corrected.

**Changes:**

• L. 73-75 (Section 1.3): Added sentence: "Digisonde drift measurements are performed in a dedicated fixed-frequency mode that does not produce ionograms during the measurement intervals and therefore do not provide information about the full ionospheric profile."

L. 104: Acknowledge Wright and Pitteway (1994).

**Response:** The citation has been added.

**Changes:**

• Line 104 (Section 2.1): Reference to Wright and Pitteway (1994) included in sentence on development of drift estimation.

L. 217: Clarify that several minutes refers to cadence, not sweep duration.

**Response:** Sentence revised.

**Changes:**

• Line 217-223: Changed to: "Historically, standard ionogram soundings were typically performed at 15-minute intervals at most ionospheric stations worldwide. Hourly values of key ionospheric parameters were then usually manually scaled and submitted to central databases. When using a Digisonde in standard operating mode, a single ionogram measurement typically takes about 1–2 minutes, depending on the specific configuration (such as the range of sounding frequencies and the frequency step). In recent years, some stations have adopted denser sounding schedules, with standard cadences of 5 minutes."

**Section 4.2 – smoothing clarification:**

**Response:** Clarified that smoothing is applied after the drift calculation.

**Changes:**

• Line 349-354 (Section 4.2): Added text: "For clarity, we note that the vertical drift values were first computed as differences of height parameters (e.g., ΔhmF2) measured at time-separated ionograms, using predefined time intervals Δt (1, 5, 15, 30, or 60 minutes). The resulting time series was then smoothed using a centered moving average. We also tested the reverse procedure—smoothing the height parameters first and then computing drift—and found no significant differences in the final smoothed time series. We therefore retained the direct-differencing approach for its simplicity and better control over temporal structure."

**Reviewer 2**

**Comment 1:**

**Reviewer:** Outline applicability to standard cadence datasets.

**Response:** We added a statement to Section 5 discussing how our approach could be extended to long-term datasets at 5-minute cadence.

**Changes in Manuscript:**

• Lines 413–415 (Section 5): Text added: "In contrast, a 5-minute step—which corresponds to the standard ionogram cadence at some stations—still frequently yields unstable and inconsistent drift estimates. Therefore, in the following discussion we focus on results derived from 15-minute averaged inputs. Even at this time scale, however, notable discrepancies between the individual methods persist in many cases."

**Comment 2:**

Reviewer: Discuss and interpret DDM standard deviations.

**Response:** Addressed. A new paragraph in Section 5 explains the variation in DDM uncertainty and its practical implications.

**Changes in Manuscript:**

• Line 270-282 (Section 4.1): Added: "Each data point is accompanied by a vertical error bar representing the standard deviation of the individual detections within each measurement window. These error bars reflect the internal variability of the detected reflection points at a given time. Notably, in some parts of the day, the uncertainties are significantly larger—especially in the raw (unsmoothed) data—while in other intervals they remain relatively small.

This variability is primarily related to the number of detected echo points during each measurement. A low number of reflections typically leads to large uncertainties due to reduced statistical confidence in the vertical drift estimate. In contrast, when many reflections are detected, the derived drift value is more stable and the standard deviation correspondingly smaller.

It is important to note that during this special measurement campaign, a very short measurements was used for both ionograms and drift measurements, resulting in reduced sounding time and fewer detected points. In contrast, during regular routine operation, each measurement is based on a significantly longer sounding sequence. Therefore, under standard measurement conditions—typically with repetition intervals of 5 to 15 minutes—substantially lower uncertainties can be expected, leading to more precise drift estimates."

**Additional Revisions:**

• All colored plots revised using colorblind-friendly palettes (Color Universal Design).

• Time series figures redesigned for clarity (thinner lines, consistent colors, improved legends).

We hope these changes meet the expectations of the reviewers and editor, and we thank you again for the opportunity to improve our manuscript.

The tracked-changes manuscript is included.

---

## Author Response (AR2)

Dear Reviewer,

We sincerely appreciate your thorough review and valuable feedback on our manuscript. Your comments have greatly helped us improve the clarity and quality of the paper. We have addressed each point in detail, and below we provide our responses together with the corresponding revisions made in the text.

**Comment 1:**

In my view, all methods are based on ionosonde data. How can one validate that the derived drifts from both methods are "correct"?

**Response:**

We agree that all investigated methods are based on ionosonde measurements. However, they represent different ways of utilizing ionogram information: (i) direct Doppler Drift Measurement (DDM) and (ii) indirect derivations from ionogram-based height parameters. The DDM provides direct Doppler frequency shifts of echoes, while the height-based approaches infer apparent drifts from temporal changes of reflection heights.

To address the issue of validation, we have clarified that our study does not aim at an exhaustive validation of absolute drift accuracy but rather at identifying internal consistency, limitations, and discrepancies among commonly used ionogram-based techniques. This clarification was added to the end of the Discussion.

**Comment 2:**

Line 55: There are other instruments that have provided vertical drifts such as IVM instrument onboard ICON.

**Response:**

A new paragraph was added acknowledging the contribution of ICON's Ion Velocity Meter (IVM) in measuring in-situ ion drifts:

"In addition, the ICON mission's Ion Velocity Meter (IVM) provides high-cadence in-situ measurements of ion drift velocity (including the vertical component near the magnetic equator) and has revealed seasonal, longitudinal and local-time variations in both vertical and zonal drifts (Heelis et al., 2017; Park et al., 2021)." (Section 1.1, lines 63–67)

**Comment 3:**

Lines 80-81: Is there a study comparing DDM drifts with other datasets? For-example, the authors reference work done over Jicamarca. There is also a digisonde in Jicamarca. Any validation studies of DDM data with Jicamarca ISR?

**Response:**

We clarified that such a study exists and explicitly referenced Woodman et al. (2006), who compared ISR and Digisonde-derived drifts over Jicamarca:

"Several studies have attempted to compare different approaches to evaluate their consistency. Woodman et al. (2006) compared vertical and zonal drift measurements obtained from the Jicamarca ISR with those derived from the Digisonde, focusing on a few case-study days. Their results showed good agreement for the vertical drifts at periods when convection dominates (e.g., during the nighttime and the pre-reversal enhancement), whereas the daytime correspondence when production and recombination are not dominant was generally poor. Yue et al. (2008) analyzed the correspondence between Fejer–Scherliess empirical model

predictions of the equatorial E×B drift and hmF2-derived variations, finding that the model and hmF2-based drifts agreed well during specific local-time intervals (around sunrise and sunset).

These comparisons illustrate that while different methods may not always agree in detail, they often preserve key temporal trends and offer complementary perspectives on ionospheric dynamics."

(Section 1.4)

**Comment 4:**

The last statement on page 3: "In this study, we aim to address this gap.... derived from ionosonde and digisonde measurements". But digisondes are just digital ionosondes and so it is the same instrument. The authors portray them as different instruments. Perhaps, it can be talked of as different measurement capabilities from one instrument?

**Response:**

We revised the wording to clearly reflect that the Digisonde is a digital ionosonde equipped with additional DDM capability:

"In this study, we aim to address this gap by comparing vertical drift estimates obtained from ionosonde data (derived from ionogram analysis) and direct drift measurements provided by the Digisonde system. Since the Digisonde is a digital ionosonde with additional drift-measurement capability (DDM), it offers an ideal platform for comparing the results of both approaches using data from the same instrument. These data will then be compared across different temporal scales, enhancing our understanding of both method-specific uncertainties and the underlying plasma dynamics."

(Section 1.4)

**Comment 5:**

Line 160: Define acronyms POLAN and ARTIST.

**Response:**

We added brief definitions for both terms when first mentioned (including references):

"A comparison of results obtained using the POLAN (POLynomial ANalysis) and ARTIST (Automatic Real-Time Ionogram Scaler with True Height) algorithms is presented in Šauli et al. (2007). Further details on POLAN can be found in Titheridge (1985), and on ARTIST in Reinisch (1996) and Reinisch et al. (2005), among other sources." (Section 2.2)

**Comment 6:**

Lines 225–235: Why do the authors expect a fair comparison when the experiments' design is different...

**Response:**

When designing high-cadence campaigns, certain trade-offs are unavoidable, such as reduced number of pulse repetitions (higher noise level), increased frequency step (coarser ionogram trace), or restriction to the ordinary mode only. Two campaigns were deliberately designed with slightly different configurations to minimize the influence of any non-standard setup on the results. Both configurations proved suitable for determining the vertical component of the apparent drift without loss of measurement quality.

**Comment 7:**

Lines 290-295: The authors should include a description about the limitations of using successive heights in estimating vertical drifts. Based on this methodology, ionosonde derived F-region drifts may not be reliable during periods of dominant production and recombination (local daytime). See Bittercourt and Abdu (1981), A theoretical comparison between apparent and real vertical ionisation drift velocities in the equatorial F region, JGR. This should be considered while interpreting results.

Essentially, ionosonde derived drifts based on this method are close to zero during local daytime and differences reach 10-20 m/s compared to ISR dataset (Woodman et al., 2006; Comparison of ionosonde and incoherent scatter drift measurements at the magnetic equator, GRL). The reason is that for a constant drift over some time, the electron density profile may near equilibrium and the virtual height will almost be the same at successive times. The derivative becomes significantly small or even zero resulting in very small drifts which may not be realistic during local daytime.

**Response:**

We expanded this part to emphasize that the  $\Delta hmF2/\Delta t$  method provides only apparent drifts and to cite Bittencourt & Abdu (1981) and Woodman et al. (2006):

"It should be emphasized that this approach provides an apparent drift, reflecting the temporal change of the F2-layer peak height rather than the true plasma motion. As discussed in Sect. 1.4 and by Bittencourt and Abdu (1981), such ionosonde-derived F-region drifts may not represent actual plasma motion during periods of dominant production and recombination (typically in local daytime). Therefore, interpretation of these results must consider these limitations when comparing with direct drift measurements." (Section 2.4)

"Essentially, ionosonde-derived drifts based on successive heights tend to approach zero during local daytime, as the ionospheric electron density profile may remain close to equilibrium despite the presence of actual plasma motion. Consequently, the apparent change in virtual or true height between consecutive ionograms becomes very small, leading to unrealistically low drift values. This limitation, already discussed by Bittencourt and Abdu (1981) and further demonstrated by Woodman et al. (2006), emphasizes that ionosonde-derived F-region drifts may not reflect the true plasma motion under strong production and recombination conditions."

(Discussion)

**Comment 8:**

In summary, while the comparisons of the two techniques estimating vertical drifts is relevant, it would have been more convincing and helpful to validate these results. For-example, a methodology of using successive heights and time can be performed over Millstone Hill which has both ISR and ionosonde.

**Response:**

We acknowledged this suggestion and clarified the scope of our study:

"The aim of this study was not to provide an exhaustive validation against all available techniques, but rather to highlight the intrinsic behavior, limitations, and potential inconsistencies of commonly used ionogram-based drift estimation methods. A detailed validation against incoherent scatter radar (ISR) data, would certainly be valuable for future work and would help to further quantify the reliability of different methods across varying

geophysical conditions." (Discussion)

**Summary**

All reviewer comments were addressed directly in the revised manuscript. The relevant clarifications and new citations - Titheridge (1985); Reinisch (1996); Heelis et al. (2017); Park et al. (2021) have been incorporated in the text.